# Phosphorus Transport in a Lowland Stream Derived from a Tracer Test with $^{32}$P

**Damian Zięba** [†] 🆔 **and Przemysław Wachniew** *,[†] 🆔

Faculty of Physics and Applied Computer Science, AGH University of Science and Technology, Mickiewicza 30, 30-059 Kraków, Poland; damian.zieba@fis.agh.edu.pl

*   Correspondence: wachniew@agh.edu.pl
†   These authors contributed equally to this work.

**Abstract:** Small streams in urbanized rural areas receive loads of P from various, often episodic, sources. This paper addresses, through a tracer test with $^{32}$P, retention and transport of a pulse input of phosphorus in a 2.6 km long stretch of a channelized second-order lowland stream. Tritiated water was introduced alongside the $^{32}$P-labelled ortophosphate in order to isolate the influence of the hydrodynamic factors on P behavior. Tracer concentrations in unfiltered water samples were measured by liquid scintillation counting. Four in-stream and five hyporheic breakthrough curves were collected at four points along the stream, two of which encompass a beaver dam impoundment. The overall retention efficiency of $^{32}$P along the studied reach was 46%. The transient storage transport model OTIS-P provided reasonable fits for in-stream breakthrough curves (BTCs) but failed at reproducing the hyporheic BTCs. The overall small effect of transient storage on solute transport was higher in the stretch with a more pronounced surface storage. Transient storage and phosphorus retention were not enhanced in the beaver dam impoundment.

**Keywords:** phosphorus; stream; transient storage; hyporheic zone; tracer test; $^{32}$P; tritium; beaver dam; OTIS





## 1. Introduction

Anthropogenic phosphorus enrichment is one of the principal factors in the deterioration of surface water quality, with rivers being an important component in human-affected phosphorus cycling. Rivers receive excessive phosphorus loads from urban and agricultural sources and carry this nutrient to lakes and coastal seas. Therefore, the understanding of phosphorus cycling and retention in streams is important for the improvement of eutrophication management tools in river catchments and in their recipient seas, which may not rely on load reduction only [1–3]. Riverine phosphorus cycling is complex and involves transformations between the inorganic-organic and dissolved-particulate forms of different bio-availability [4,5]. Furthermore, the riverine ecosystems and their internal phosphorus processing are constrained and controlled by hydrological and hydromorphological factors [6]. The coupled action of biotic and abiotic phosphorus cycling and downstream transport of dissolved and particulate phosphorus forms is conceptualized as spiralling, with the spiralling length reflecting the efficiency of nutrient utilization in stream ecosystems [7–9].

Due to the complexity of riverine phosphorus cycling the quantification of uptake rates and lengths is not straightforward and is best addressed through phosphorus addition experiments. Tracing of intentionally introduced substances (artificial tracers) at field and laboratory scales in order to understand the functioning and status of hydrological systems is an essential tool in water resources management [10–14]. In this regard, whole-stream additions of radioactive phosphorus present advantages in quantification of phosphorus cycling over stable phosphorus additions and batch experiments [15,16]. Phosphorus has

two radioactive isotopes ($^{32}$P, $T_{1/2}$ = 14.268 d; $^{33}$P, $T_{1/2}$ = 25.34 d) manufactured artificially by nuclear transmutation and used widely as radiopharmaceuticals and tracers in biochemical and biomolecular research. Both isotopes are also naturally produced in the atmosphere through interaction of cosmic radiation with argon nuclei [17,18], but their applications as the environmental tracers are limited to studies of atmospheric and oceanic mixing. Artificially produced $^{32}$P and $^{33}$P are used at field- and laboratory-scale to trace and quantify phosphorus uptake, speciation and transport in ecosystems and soils [19–21]. Applications related to the aquatic environment were initially concerned with the pathways of phosphorus uptake by biota in lakes [22,23] and rivers [24–26]. Results of tracer tests with radio-phosphorus were instrumental in the identification of the key features of phosphorus cycling in streams, such as uptake rates [7,8,27,28], food-web pathways [29] and the influence of transient storage [30]. An important advantage of radioactive phosphorus injections, comparing to stable phosphorus additions, is that they introduce a negligible amount of phosphorus, which does not affect stream uptake capacity. Despite the high relevance of radio-phosphorus tracing to deconvolute phosphorus cycling in rivers the application of this method at field scale is hindered, and in many countries practically impossible, due to the restrictions imposed by radiation protection regulations. Consequently, methods based on stable phosphorus addition to quantify the relative contribution of abiotic and biotic processes [16] and to quantify uptake lengths [31] have been refined [16,31] and methods based on tracing stable isotopic composition of oxygen in phosphate [32–34] have been developed.

Phosphorus retention in small streams in urbanized rural areas is an important, and probably globally relevant, but not well quantified aspect of phosphorus cycling in catchments. These streams receive significant loads of phosphorus from various, often episodic, sources that collectively constitute a semi-diffuse source not accounted for in pollution assessments [1,35,36]. Among such sources are farmyard, urban and road runoff, sewage and wastewater disposal, fish ponds. In rural Poland, where only 42% of the population is connected to sewerage systems, the improper wastewater management may be a significant source of nutrient pollution [37]. Wastewater effluent phosphorus may be effectively sequestered in stream biota and sediment and remobilized under storm flows [1,16]. However, large inputs of nutrients, whether continuous or episodic, may exceed the retention capacity of streams [38–40].

This work presents and discusses results of a tracer test with $^{32}$P performed to characterize phosphorus retention and transport in a 2.6 km long stretch of a second-order rural stream subjected to multiple sources of phosphorus contamination. Tritium as tritiated water was injected simultaneously with $^{32}$P-labelled ortophosphate in order to assess in-stream transport and transient storage exchange characteristics of a conservative tracer. Previous tracer tests with $^{32}$P which aimed at quantification of phosphorus uptake rates/lengths in streams [7,8,27,28,30] were performed in short (<200 m) reaches of streams characterized by low mean discharges (<20 L/s), with tracers released at constant rates during 30–60 min periods. In comparison, the instantaneous injection performed in this study was traced during 6 days along the 2.6 km long stream stretch at the discharges varying between 100–200 L/s. While most phosphorus addition experiments aimed at quantification of phosphorus cycling in streams were performed with constant injection rates, the instantaneous injections provide equally reliable estimates and may be more appropriate at variable backgrounds [41], which might be a common feature of small streams in urbanized rural areas. Furthermore, pulse injections are more adequate for the analysis of long-term behavior of conservative solutes as affected by the exchange with the hyporheic and other transient storage zones [42,43]. Finally, the pulse injection of radioactive tracers simplifies the injection procedure and reduces exposure times of the personnel to ionizing radiation.

The main goal of this work is to characterize, in the timescale of the carried out tracer test (6 days), three subsequent stretches of the stream with respect to: (1) overall phosphorus retention capacity and (2) influence of transient storage on phosphorus retention

and transport. The first goal is achieved by a comparison of tracer recoveries between the sampling points. The influence of transient storage is assessed through the metrics derived from parameters of the transient storage transport model OTIS-P [44]. The results of this study contribute to the understanding of the fate of the episodic inputs of inorganic phosphorus in channelized streams during the non-growing season. The relative importance of such semi-diffuse sources of phosphorus to small streams is expected to increase due to urban sprawl [45]. On the other hand, in the Northern Hemisphere the degraded stream ecosystems are now impacted by the expansion of beavers [46,47]. Accordingly, the presented study addresses the influence of a beaver dam impoundment on phosphorus retention and transport.

## 2. Materials And Methods

### 2.1. Study Site

The Kocinka (length 40.2 km; catchment area 257.8 km$^2$; average discharge at the outlet 1 m$^3$/s) is a lowland river in Southern Poland, in the Baltic Sea basin. The river is polluted by nitrate of agricultural and wastewater origin and as such was a subject of an interdisciplinary study concerned with the development of novel methods for the governance of agricultural water pollution [48,49]. The river is groundwater-fed [50] with the streamflow and water quality being controlled by nitrate-polluted groundwaters discharging from a large fissured-karstic aquifer in Jurassic limestones and, in the headwater part, from shallow porous aquifers in Quaternary sands and gravels of fluvioglacial origin. Total dissolved phosphorus (TDP, 0.45 μm filtered) concentrations were observed occasionally in the period 2015–2017 at two sampling points encompassing the stream stretch studied in this tracer test (Figure 1). Mean TDP concentrations at the upstream and downstream points were, respectively 1.810 mg/L and 0.948 mg/L. The high and variable (within one order of magnitude) TDP concentrations reflected domestic wastewater inputs from the partly urbanized catchment, including incidental sewer overflows and dumping of untreated sewage. A consistent drop in TDP between the two sampling points might be, at least partly, explained by dilution by lateral inflows received through ditches. The stream may also receive groundwater discharges directly through the bed, as evidenced in the more downstream parts of the Kocinka [50]. Indeed, stream discharge showed a significant downstream increase during the experiment (Table 1).

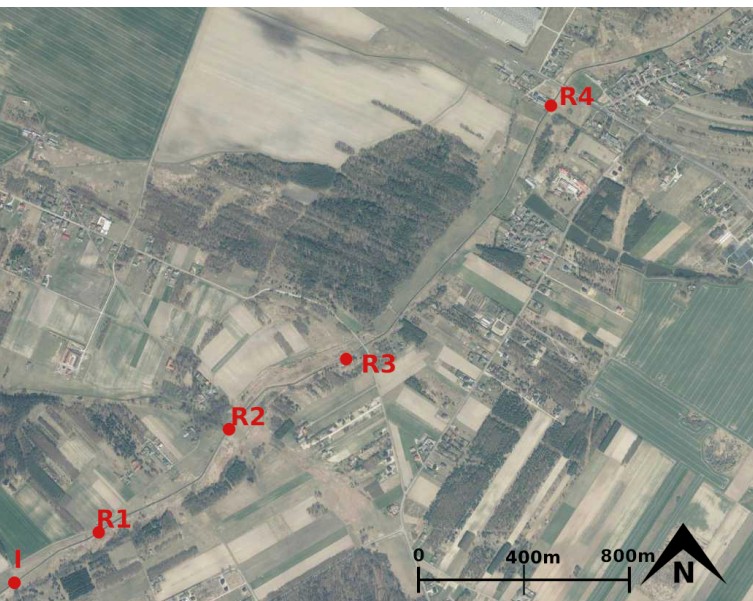

**Figure 1.** Aerial picture of the study site taken before the construction of the beaver dam. Dots mark injection point I and sampling points R1–R4. Modified from https://mapy.geoportal.gov.pl/ (accessed on 15 April 2017).

**Table 1.** Stream discharge during the tracer test [m³/s].

| Point | 5 April | 7 April |
| --- | --- | --- |
| R1 | 0.142 | 0.095 |
| R2 | - | - |
| R3 | - | 0.111 |
| R4 | 0.202 | 0.147 |

The artificially straightened stream channel is incised in a flat and wide valley bottom covered by meadows and wetlands (Figure 2). Sampling points were selected to encompass river stretches with different hydromorphological conditions (Figure 1). The distance between the injection point I and the first sampling point R1 (305 m) was large enough to ensure good transversal mixing of tracers, according to even the most conservative estimates [51,52]. Stream channel in stretches R1–R2 and R3–R4 has regular, parallel edges with the width and cross-section area fluctuating around 3.5 m and 1.3 m², respectively. The R2–R3 stretch overlaps with a backwater zone upstream a beaver dam. Channel width and cross section area increase gradually along this stretch as water flow is slowed down. Besides the beaver dam the longitudinal profile of the stream is affected by several small concrete weirs and artificial riffles. During the growing season the entirely unshaded stream channel is overgrown with emergent plants that significantly reduce water velocity and open channel width. During the experiment, stream channel was mostly open, with flow blocked by dead vegetation only along stream banks (Figure 2).

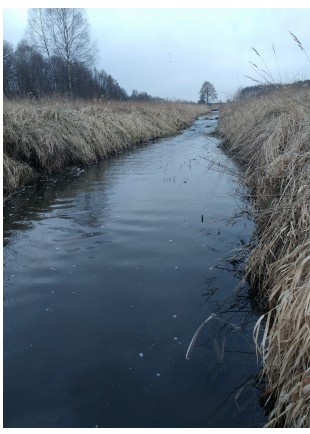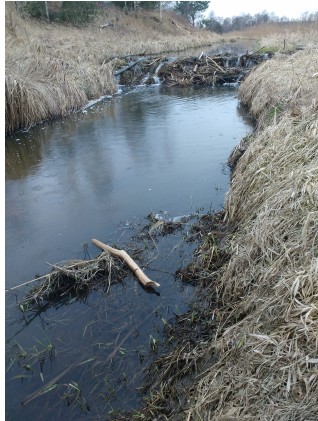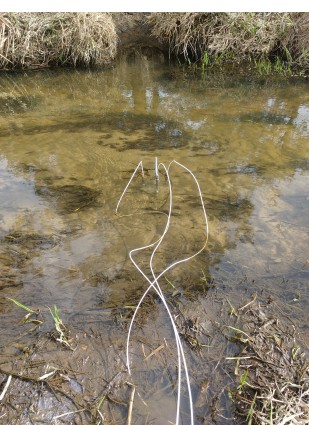

**Figure 2.** From the left: stretch R1–R2; beaver dam—picture taken looking upstream from point R3; hyporheic water sampler at point R4.

*2.2. Tracer Test Design*

The usefulness of a tracer test in characterization of solute transport in rivers depends on the quality of tracer breakthrough curves (BTCs). The reliability of the estimates of different transport parameters depends on how well particular regions of a BTC are represented by sampling [53]. Additionally, given the short half-life of ³²P (14.268 days) the application of this tracer requires a careful design of sampling and analysis schemes. In order to reduce uncertainties in the analysis of low activity samples their measurements have to be performed as quickly as possible after sampling. Therefore the time frames for the passage of the low activity parts of the BTCs at particular sampling locations need to be predicted before tracer injection. Application of auxiliary dye tracers for tracing of the injected plume was inappropriate because of possible interferences in liquid scintillation counting. Therefore, the conservative tracer BTCs were predicted for the sampling points by the OTIS-P model (without transient storage) using discharge and water velocity values measured on the day preceding tracer injection and the dispersion coefficient derived from an independent tracer experiment. The BTCs obtained during the nitrate addition

experiment performed in October 2017 in the stretch I-R1 ($Q$ = 123 L/s) were used to estimate the dispersion coefficient. Based on these predictions, sampling schedule was designed for the period of 72 h after the injection. Sampling was extended to 147 h because samples from the tail regions of BTCs collected at points R3 and R4 appeared to have measurable $^{32}$P activities.

Tracer injection was performed on 5 April 2018, 1 pm at point I (Figure 1). The injected activities were estimated by the supplier as 30 GBq of $^3$H (tritiated water) and 10 GBq of $^{32}$P (aqueous solution of phosphoric acid) with 10% relative uncertainties. Potential doses of ionising radiation to the personnel, general public and aquatic organisms were estimated as not harmful and approved by the Polish Atomic Agency through their permit to introduce the above activities of both tracers into the Kocinka River. Tracers were delivered to the injection site by a licensed transport company. The glass ampules with tracers were crushed in a plastic container filled with 5 L of water. Contents of the container were emptied in a riffle area of the stream, which facilitated cross-sectional mixing of tracers. The weather was stable during the experiment, with no precipitation and air temperatures fluctuating between $-2$ to $+ 10\,^\circ$C.

### 2.3. Tracer Sampling and Analysis

Water samples were collected from the central part of the stream channel by a plastic bottle attached to a sampling pole. Samples were immediately transferred to 22 mL HDPE liquid scintillation vials containing 0.1 mL of 0.1 N of $H_3PO_4$ to prevent tracer adsorption on vial surfaces. Dilution by the added acid volume was taken into account in calculations of $^{32}$P specific activities. Sample activities were later measured in the same vials. Contrary to the previous $^{32}$P tracer studies conducted in streams our water samples were not filtered, primarily because this study was not concerned with the quantification of phosphorus pathways and pools in stream ecosystem. The main objective of this work was to assess an overall effect of the abiotic and biotic processes on the retention and transport of phosphorus in the stream. It must be also noted that filtration through the commonly used 0.45 μm membranes does not separate the finest particulates and the filtering process itself may affect dissolved P concentration in the filtrate [54]. Indeed, tests performed on 35 stream water samples showed a large variability (5%–76%) in the fraction of $^{32}$P lost during filtration. Furthermore, a large fraction of phosphorus in surface waters is associated with iron-rich colloids [55,56] most of which pass through the 0.45 μm membranes. The truly dissolved, easily bioavailable ortophosphate ion is probably not the dominant movable form of phosphorus in rivers.

At points R3 and R4 water samples were collected also from the hyporheic zone by use of a dedicated sampler consisting of an array of stainless steel tubings pushed into bottom sediment. Each of the tubings had at its lower end two orifices of 0.1 mm diameter through which pore water was sucked out. Samples were collected from the depths of 5, 10 and 15 cm in sediment, however, at site R4 water could not be retrieved from the depth of 15 cm. Pore water samples were slowly sucked into syringes connected to the sampler through a 1.5 m long small diameter teflon tubing. Before sampling the dead volume of the teflon tubing was removed. Water collected in syringes was immediately transfered to vials. Each syringe was used only once to avoid cross-contamination of samples. The sampling arrays remained in sediment between samplings, however they were removed after the first 72 h and placed in the same locations for the extended part of the experiment.

Activity concentrations of $^3$H and $^{32}$P in stream water samples were measured by liquid scintillation counting method with the Quantulus 1220 liquid scintillator beta/alpha spectrometer. Two different scintillators were used for $^{32}$P (Insta-Gel Plus) and $^3$H (Ultima Gold LLT) measurements. The ratios of scintillator to water sample volumes optimized to obtain the highest efficiency of the detection were 10 mL/10 mL for $^{32}$P and 12 mL/8 mL for $^3$H. After scintillators were added the vials were kept in the apparatus for 24 h before the measurement in order to reduce chemiluminescence. Measurements were performed for batches containing 18 samples, a reference activity solution and a blank sample. The

activities of the reference solutions were used to determine counting efficiency for each batch of samples. Dead water was used as the $^3$H blank and stream water collected before tracer injection was used as the $^{32}$P blank. Therefore, the measured $^3$H activities include the environmental component. Activity concentrations of the environmental $^3$H measured in the Kocinka at point I in the years 2015–2016 varied from $7.2 \times 10^{-4}$ kBq/L to $8.6 \times 10^{-4}$ kBq/L. Due to the partial overlap of the $^3$H and $^{32}$P spectra the activity of each sample was measured twice. The first series of measurements was performed within 10 weeks after collection of samples, beginning with samples with the lowest expected activities. Measurements performed for $^3$H and $^{32}$P reference solutions allowed to identify the spectral overlap range and to evaluate a correction factor. The proportion of $^{32}$P counts in the overlap range was 5% of total counts. The second series of measurements began 4 months later, after the $^{32}$P activities of samples decreased to around 0.3% of the initial values and practically only $^3$H was present in the samples, which improved the accuracy of $^3$H activity determination. The measured activity concentrations of the tracers are presented in Appendix A.

*2.4. Data Treatment and Otis-P Modeling*

Because sampling extended for 43% of $^{32}$P half-life, which corresponds to the radioactive decay of 26% of the initial activity, measured activities were corrected for the decay. While the transport model predicts the actual concentrations (specific activities) of the tracer at the moment of sampling, the estimates of tracer recovery at sampling points have to be based on the activities at the time of injection. Accordingly, the measured $^{32}$P activities were recalculated for each sample for the moments of sampling and injection.

For the estimation of tracer recovery and of the retention and transport metrics the discharge, water velocity and channel dimensions have to be estimated. Particularly, the OTIS-P model requires quantification of discharge, lateral inflows and channel cross-section areas along the stream. Channel widths and cross-section areas were measured on the day before tracer injection at 20 cross-sections roughly evenly distributed between points I and R4. Discharge was measured a few hours before tracer injection at points R1 and R4 and 2 days after the injection at points R1, R3 and R4 (Table 1). Discharge was calculated by the area-averaged integration of water velocities measured by a 801 Valeport electromagnetic velocity meter. While discharge increased downstream due to lateral inflows, it also significantly decreased at each point during the course of the experiment. The input data for OTIS-P model were interpolated for different cross sections and times based on the measured discharges, assuming that discharges did not change after the second round of measurements performed on 7 April. The BTCs obtained at each point were treated as tracer input functions for subsequent stretches. Parameters of the transient storage model OTIS-P were then fitted to in-stream and hyporheic BTCs obtained at points R2–R4. The BTCs were integrated using the trapezoidal rule, taking into account variations of the discharge interpolated in time and space, to obtain total activities of tracers recovered at sampling points (Table 2).

Figure A1 presents processes and the corresponding model parameters used in OTIS-P to describe the advective-dispersive-reactive transport of solutes in the main channel of a stream, exchange of solutes between the main channel and the storage zones, and the reactive processes in the latter [44]. Model parameters were fitted stretch-wise treating the BTCs obtained at each sampling point as tracer input functions for subsequent stretches. The model was run at the spatial resolution of 1 m and the temporal resolution of 0.01 h. The downstream increase of discharge was implemented as lateral inflow, with a uniform distribution along each stretch. Tracer concentrations in the lateral inflow were assumed to be $8.6 \times 10^{-4}$ kBq/L for $^3$H and 0 Bq/L for $^{32}$P. Parameters of the conservative tracer model ($D$, $A$, $A_s$, $\alpha$) were fitted automatically to main channel BTCs, assuming constant discharge, by the use of the Nonlinear Least Squares technique implemented in OTIS-P with weights emphasizing the tail concentrations [44,57]. The $A$ values were in the next step adjusted manually to account for the observed spatially and temporary variable discharge. The final

conservative tracer model values of $D$, $A$, $A_s$ and $\alpha$ were sought by trial-and-error approach to obtain satisfactory fits to the hyporheic $^3$H BTCs. The above values were used as the initial values for the $^{32}$P model in which, again, automatic fitting was used for the main channel BTCs and trial-and-error fitting for the hyporheic BTCs.

**Table 2.** Recovered tracer activities (corrected to the injection time) and $^{32}$P retention at sampling points. Distance is measured from the injection point I.

| Point | Distance [m] | Recovered $^{32}$P [GBq] | Recovered $^3$H [GBq] | $^{32}$P/$^3$H | $^{32}$P Recovery Relative to R1 | $^{32}$P Retention Relative to R1 |
|-------|-------------|--------------------------|------------------------|----------------|----------------------------------|-----------------------------------|
| R1 | 305 | 10.86 | 37.3 | 0.29 | 1 | 0 |
| R2 | 970 | 8.84 | 37.9 | 0.23 | 0.80 | 0.20 |
| R3 | 1540 | 6.74 | 37.5 | 0.18 | 0.62 | 0.38 |
| R4 | 2594 | 5.98 | 37.9 | 0.16 | 0.54 | 0.46 |

## 3. Results and Discussion

### 3.1. Whole-Stretch Retention of Phosphorus

Table 2 presents tracer activities recovered at subsequent sampling points. The BTCs were integrated using the trapezoidal rule, taking into account variations of the discharge interpolated in time and space, to obtain total activities of tracers recovered at sampling points (Table 2). Assuming that tritium behaved like a conservative tracer, the downstream decrease of the ratio of recovered activities (the fifth column) reflected the gradual retention of $^{32}$P along the stream. Because of the large uncertainty of the injected activities these ratios were recalculated relative to activities recovered at point R1 (the sixth column). Finally, the last column presents fractions of $^{32}$P retained in the subsequent stretches of the stream relative to point R1.

Table 3 presents metrics of $^{32}$P tracer retention for the three subsequent stretches and for the whole studied stretch. Assuming that tracer removal rate along the stream is proportional to its amount remaining in water, the activities that pass through subsequent stream cross-sections must decrease exponentially with the distance [8]. The uptake length $S_w$ was thus calculated as:

$$S_w = \frac{x_j - x_i}{ln(A_i/A_j)}, \tag{1}$$

where $x$ is the distance of a cross section from the injection point ($j$ being further downstream than $i$) and $A$ is total tracer activity that passed the cross section. The first-order uptake coefficient $\lambda_r$:

$$\lambda_r = \frac{v}{S_w}, \tag{2}$$

where $v$ is water velocity, reflects the overall uptake capacity of the stream, including both the in-stream and transient storage uptake of the tracer through abiotic and biotic processes. It is not equivalent to either of uptake coefficients of the OTIS-P model, which describe separately uptake in the main channel and the storage zone [44]. The $\lambda_r$ values calculated from Equation (2) were corrected for radioactive decay by subtracting the decay constant of $^{32}$P ($\lambda_{32P} = 5.6 \times 10^{-7}$ [1/s]). The uptake velocity $v_f$ was used to compare retention capacity at varying stream depth [58]:

$$v_f = \frac{Q}{W \times S_w}, \tag{3}$$

where $Q$ is discharge and $W$ is stream width.

**Table 3.** Metrics of the movable phosphorus uptake between sampling points.

| Stretch | $v$ [m/s] Velocity | $W$ [m] Width | $Q$ [m³/s] Discharge | $S_w$ [m] Uptake Length | $\lambda_r$ [1/s] Uptake Coeff. | $v_f$ [mm/min] Uptake Velocity |
|---------|--------------------|---------------|----------------------|-------------------------|----------------------------------|--------------------------------|
| R1–R2 | 0.116 | 3.5 | 0.132 | 2931 | $3.6 \times 10^{-5}$ | 0.75 |
| R2–R3 | 0.049 | 4.7 | 0.133 | 2225 | $1.8 \times 10^{-5}$ | 0.78 |
| R3–R4 | 0.152 | 3.5 | 0.166 | 8130 | $1.5 \times 10^{-5}$ | 0.35 |
| R1-R4 | 0.094 | 3.9 | 0.166 | 3736 | $2.1 \times 10^{-5}$ | 0.68 |

The shortest uptake length $S_w$, found for the beaver dam impoundment, did not translate into the highest uptake velocity $v_f$, which had nearly the same value for stretches R1–R2 and R2–R3. Apparently, the sedimentation of coarser particulates that might be expected to occur in the impoundment [59] did not affect the movable fraction of $^{32}$P. The stretch R3–R4, where water velocity was the highest among the three stretches, had a significantly lower uptake velocity.

The values of uptake velocity $v_f$ obtained in this study (Table 3) fall in the lower range of the estimates derived from stable phosphorus additions [16,41,60–62]. Spiralling lengths derived from results of $^{32}$P tracer tests for two contrasting streams with small and large volumes of transient storage zone were estimated at 643 m and 111 m, respectively [30]. Based on the data provided in [30] the respective uptake velocities could be calculated at 0.18 mm/min and 1.07 mm/min. A direct comparison of $v_f$ estimates obtained for rivers of different size, hydraulics, bed substrate and P loads and for different modes of P additions (pulse-continuous, stable-radioactive) was of limited value. Nevertheless, the consistency of our results with other studies indicated that tracing of all movable P fractions, without water sample filtration, did not bias the estimates of P retention metrics. Since colloidal organic and inorganic particles, majority of which is 0.45 μm filterable, may be the main carrier for phosphorus transport in streams [55,63] the unfiltered stream water samples properly represented the movement of a reactive tracer plume.

*3.2. Characteristics of $^3$H Transport*

Figures 3–6 present semi-log plots of the observed and modeled BTCs for points R2–R4. Linear plots underemphasized the tail regions of the BTCs and made them indistinguishable. The OTIS-P model was capable to fit reasonably well the in-stream $^3$H BTCs for all sampling points, but the hyporheic BTCs could not be satisfactorily reproduced. At point R3 (Figure 3) the model overestimated the hyporheic $^3$H for the first 2 days (48 h) of the experiment and underestimated it for the later period. The spatial distribution of $^3$H in sediment observed at R3 during the first 2 days ($C_{5cm} < C_{15cm} < C_{10cm}$) is the result of competition between two effects that the depth in sediment has on hyporheic exchange: the hydrodynamically induced flux decreases with depth [64], while the hyporheic travel times are longer for the deeper penetrating flow paths. The apparent lag times between the in-stream and hyporheic BTCs at different depths arose due to not only delayed tracer penetration into sediment but also due to tracer dilution with the pre-experiment interstitial water. Therefore, the 15 cm BTC seemed to lag behind the stream BTC less than the 10 cm BTC. The same model explained the convergence of concentrations at 10 cm and 15 cm after the first 2 days and the excess of $^3$H at 15 cm seen after 60 h. The linear plot inserted in Figure 3 confirms the above interpretation by showing the fastest response of the 5 cm BTC to in-stream tracer variations and the decrease of concentrations at 15 cm relative to 10 cm. These observations clearly confirmed that the well-mixed transient storage zone of the OTIS-P model did not properly represent actual distribution of tracers in the hyporheic zone [64,65]. Similar relationships between the 5 cm and 10 cm BTCs can be seen at point R4 (Figure 4). In this case, however, the hyporheic concentrations increased between hours 60 to 120. Only a tentative explanation can be provided for the stabilization and increase of hyporheic concentrations seen during the second half of the experiment at R3 and R4. The decrease in stream discharge observed during the experiment, through its influence

on stream water velocity and depth, could have resulted in the reduction of hyporheic exchange. The reduction of water flux into sediment might have retarded flushing of tracer from the hyporheic zone at R3 and facilitated release of the delayed tracer plume from the less-connected regions of the hyporheic zone at R4.

Table 4 presents parameters of the OTIS-P model obtained through the inverse modeling ($\alpha$—storage zone exchange coefficient, $A$—cross sectional area of the main channel, $A_s$—cross-sectional area of the storage zone, $D$—dispersion coefficient [44]) as well various metrics used to characterize transient storage, derived from the above model parameters and the advective velocity $u$. Of these metrics only $F_{med}^{200}$ is based on all three quantities ($\alpha$, $A_s$, $v$) that influence the effect of transient storage on tracer transport [66]. The $F_{med}^{200}$—the fraction of median travel time due to transient storage normalized to the 200 m long steam stretch—was evaluated as:

$$F_{med}^{200} \cong \left(1 - e^{-L(\alpha/u)}\right) \times \frac{A_S}{A + A_S},\tag{4}$$

where $L$ = 200 m.

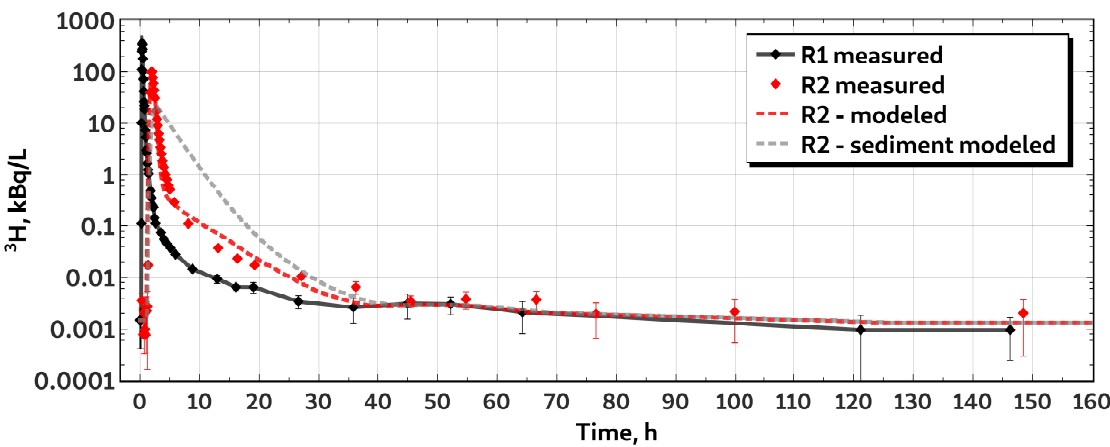

**Figure 3.** The $^3$H activity concentrations observed at R1 and R2, and modeled at R2. Observations in sediment were not performed at R2.

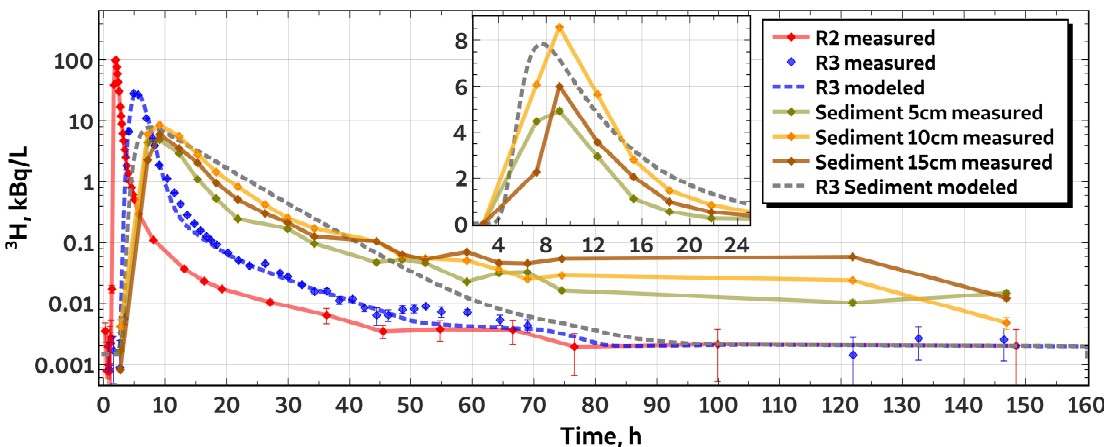

**Figure 4.** The $^3$H activity concentrations observed at R2 and R3, and modeled at R3.

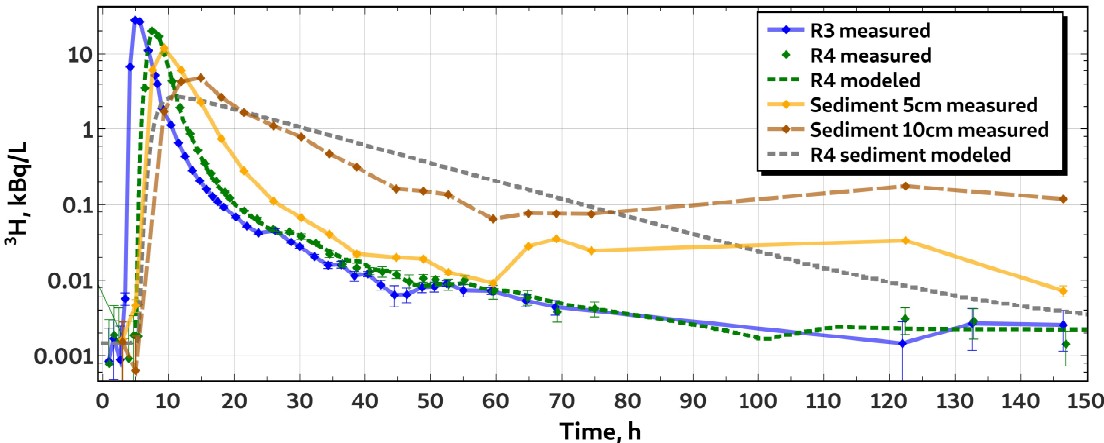

**Figure 5.** The $^3$H activity concentrations observed at R3 and R4, and modeled at R4.

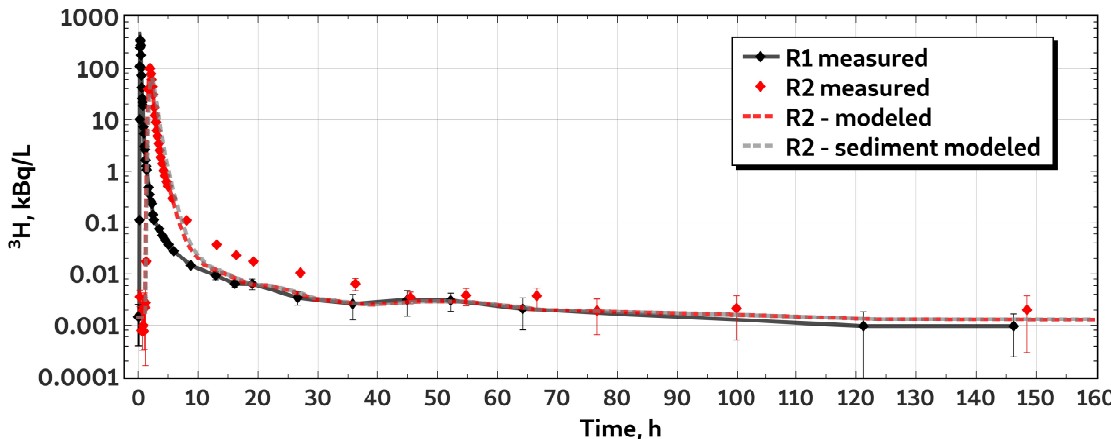

**Figure 6.** The $^3$H activity concentrations observed at R1 and R2, and modeled at R2. Values of model parameters: $\alpha = 3.2 \times 10^{-5}$ [1/s], $A_s = 0.099$ [m$^2$].

Values of the transient storage zone exchange coefficient $\alpha$ and of the metrics of the transient storage presented in Table 4 are relatively low, indicative of a small influence of transient storage on conservative solute transport. Our values of $\alpha$ are low, but still within the range reported by a meta-analysis based on results of 67 papers [67]. The modeled BTCs obtained for higher $\alpha$ did not reproduced well the observed concentrations underestimating them in the tail region. The attempts to fit higher $\alpha$ values did not significantly affect the $A_s$ values. Figure 5 shows an example of an alternative fit for point R2. On the other hand, lower $\alpha$ values led to the narrowing of BTC peaks and to the elevated concentrations in the tail. Our results showed that the $\alpha$ values can be overestimated when too much weight is placed on fitting of the peak of the BTC or when the tail region is not well represented by the measurements.

Transient storage in the studied stream may be associated with the hyporheic zone, narrow strips of dead vegetation extending along the otherwise straight and regular banks, patches of submerged vegetation and debris accumulations on the streambed. The metrics of transient storage reflected the overall small effect of these transient storage zones. While the metrics related to travel times in the main channel and storage zone as well as their sum ($t_{mean}^m$, $t_{mean}^s$, $t_{mean}$) were largest for the beaver dam impoundment (R2–R3), the $F_{med}^{200}$ values indicated that transient storage accounted for the largest fraction of total travel time in stretch R1–R2 and for the smallest fraction in stretch R3–R4. Apparently, effects of transient storage on fluxes of conservative and non-conservative solutes were not enhanced in the beaver dam impoundment. Stretch R3–R4 differed from the more upstream part of the

stream by higher water velocity, coarser bottom sediments and less frequent occurrence of submerged vegetation. The higher $F_{med}^{200}$ values between R1 to R3 reflected a higher role of surface storage comparing to R3–R4, where the hyporheic storage became relatively more significant.

**Table 4.** Parameters and metrics of transient storage derived from $^3$H BTCs. See Appendix C for explanation of symbols.

| Parameter/Metric | Unit | R1–R2 | R2–R3 | R3–R4 |
|---|---|---|---|---|
| $L$ | [m] | 665 | 570 | 1054 |
| $A$ | [m$^2$] | 1.23 | 3.18 | 1.30 |
| $Q$ | [m$^3$/s] | 0.132 | 0.133 | 0.166 |
| $u = Q/A$ | [m/s] | 0.107 | 0.042 | 0.127 |
| $D$ | [m$^2$/s] | 0.989 | 0.882 | 0.850 |
| $\alpha$ | [1/s] | $4.61 \times 10^{-6}$ | $1.11 \times 10^{-6}$ | $8.94 \times 10^{-7}$ |
| $A_s$ | [m$^2$] | 0.098 | 0.120 | 0.074 |
| $A_s/(A + A_s)$ | [-] | 0.074 | 0.036 | 0.054 |
| $P_D$ | [-] | 0.014 | 0.037 | 0.006 |
| $d_S$ | [m] | 0.049 | 0.030 | 0.037 |
| $L_s = u/\alpha$ | [m] | $2.51 \times 10^4$ | $4.37 \times 10^4$ | $1.70 \times 10^5$ |
| $T_{sto} = A_s/(\alpha A)$ | [h] | 4.8 | 9.5 | 17.7 |
| $T_{str} = 1/\alpha$ | [h] | 60 | 250 | 311 |
| $q_s = \alpha \times A$ | [m$^2$/s] | $5.66 \times 10^{-6}$ | $3.53 \times 10^{-6}$ | $1.17 \times 10^{-6}$ |
| $R_h = T_{sto}/L_s$ | [s/m] | 0.691 | 0.778 | 0.376 |
| $t_{mean}^m = Lu + 2D/u^2$ | [h] | 1.64 | 3.47 | 1.95 |
| $t_{mean}^s = (A_S/A) \times t_{mean}^m$ | [h] | 0.16 | 0.42 | 0.14 |
| $t_{mean} = t_{mean}^m + t_{mean}^s$ | [h] | 1.80 | 3.89 | 2.10 |
| $F_{med}^{200}$ | [−] | $6.33 \times 10^{-4}$ | $1.91 \times 10^{-4}$ | $7.56 \times 10^{-5}$ |

### 3.3. Characteristics of $^{32}$P Transport

Figures 7–9 present semi-log plots of the observed and modeled BTCs for points R2–R4. As for tritium, the OTIS-P model was capable to fit reasonably well the in-stream $^{32}$P BTCs for all sampling points. Differently than for $^3$H the highest hyporheic concentrations of $^{32}$P occurred at 5 cm depth. Only the 5 cm BTCs showed a roughly monotonic decay after peak concentrations which could be reasonably reproduced by the model for R4, except for the three highest concentrations. Concentrations at 10 and 15 cm fluctuated showing at R3 distinct minima during the second 24 h of the experiment. Transient storage of $^{32}$P in the hyporheic zone occurred mostly in the top layer of sediment, which, when compared with the deeper penetration of $^3$H, reflected filtration of particulate P and/or adsorption of dissolved P. The irregular behavior of $^{32}$P at greater depths might have been partly related to the changing patterns of hyporheic flow, as discussed above for $^3$H.

A comparison of OTIS-P parameters and various metrics of transient storage presented in Table 4 for $^3$H and in Table 5 for $^{32}$P shows similar values of parameters and metrics connected with tracer transport in the main channel ($A$, $u$, $D$, $P_D$, $t_{mean}^m$). This provides more evidence that non-filtered water samples adequately represented in-stream transport of $^{32}$P. Additionally, the storage zone exchange coefficient $\alpha$ was similar for both tracers, except for R1–R2 where it was almost twice higher for $^{32}$P. Consequently, $L_s$—the average distance a molecule travels downstream within the main channel prior to entering the storage zone—was roughly twice lower for $^{32}$P in R1–R2. The higher than in other stretches capacity of transient storage zone in R1–R2 to capture tracers from the main channel

was additionally enhanced for $^{32}$P. As could be expected the parameters and metrics characterizing transient storage were higher for reactive $^{32}$P than for non-reactive $^{3}$H. Interestingly, the $F_{med}^{200}$ values obtained from $^{32}$P and $^{3}$H BTCs varied between stretches almost proportionally, the former being 6.5–6.9 times higher. The effect of transient storage on $^{32}$P flux was thus proportional to the median of the travel time it spent in storage zones. The $F_{med}^{200}$ metric correlated well ($R^2 = 0.997$) with the effective uptake coefficient $\lambda_r$ (Equation (2), Table 3). This correlation confirmed that the enhanced $^{32}$P retention in R1–R2 was not due to a higher retention capacity of sediments or periphyton, nor to filtering of the coarser $^{32}$P-bearing particles but was related to the fraction of median travel time due to storage being the highest in that stretch. A direct comparison between the uptake coefficient $\lambda_r$ derived from overall tracer recoveries (Table 3) and the first order uptake coefficient $\lambda$ obtained by inverse modeling of $^{32}$P BTCs (Table 5) was meaningless because $\lambda_r$ represented the apparent uptake in the timescale of the experiment, not distinguishing between the unidirectional uptake (represented in OTIS-P by $\lambda$ and $\lambda_s$) and the reversible sorption process. Nevertheless, the lambdas derived by both methods differed for the same stretches by not more than a factor of 2.

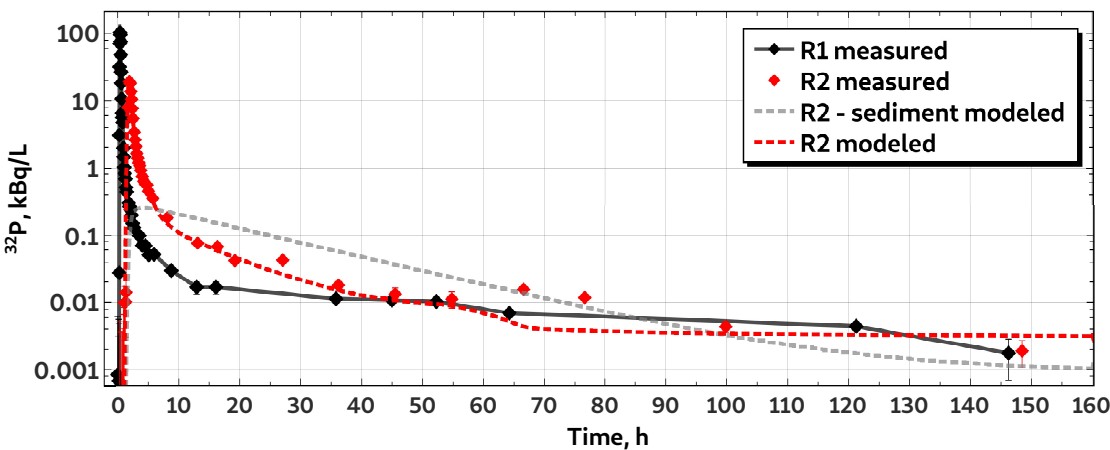

**Figure 7.** The $^{32}$P activity concentrations observed at R1 and R2, and modeled at R2. Observations in sediment were not performed at R2.

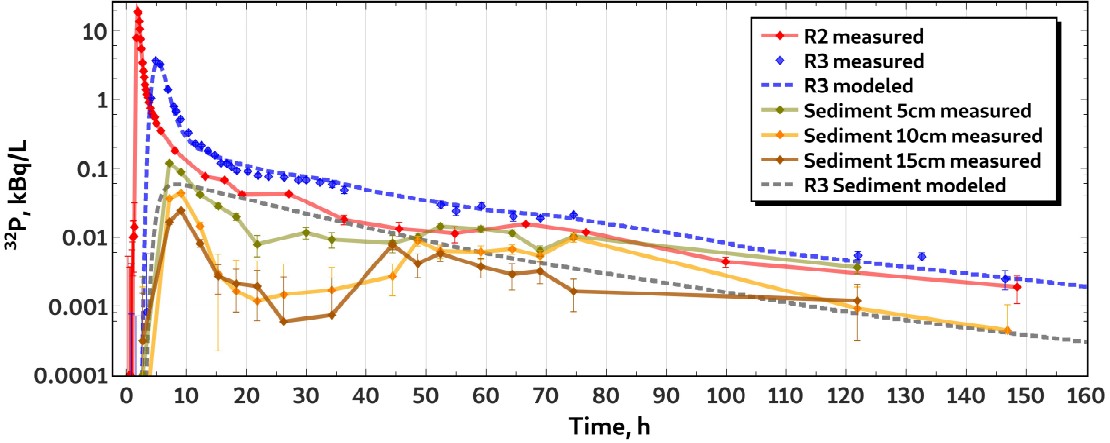

**Figure 8.** The $^{32}$P activity concentrations observed at R2 and R3, and modeled at R3.

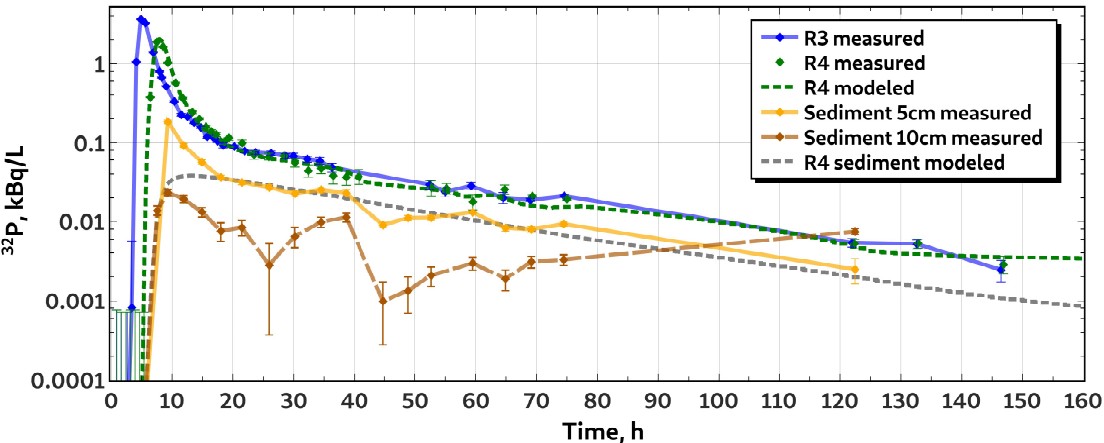

**Figure 9.** The $^{32}$P activity concentrations observed at R3 and R4, and modeled at R4.

**Table 5.** Parameters and metrics of transient storage derived from $^{32}$P breakthrough curves (BTCs). See Appendix C for explanation of symbols.

| Parameter/Metric | Unit | R1–R2 | R2–R3 | R3–R4 |
|:---:|:---:|:---:|:---:|:---:|
| $L$ | [m] | 665 | 570 | 1054 |
| $A$ | [m$^2$] | 1.213 | 3.22 | 1.40 |
| $Q$ | [m$^3$/s] | 0.132 | 0.133 | 0.166 |
| $u = Q/A$ | [m/s] | 0.109 | 0.041 | 0.119 |
| $D$ | [m$^2$/s] | 0.926 | 0.900 | 0.830 |
| $\alpha$ | [1/s] | $8.76 \times 10^{-6}$ | $9.27 \times 10^{-7}$ | $9.40 \times 10^{-7}$ |
| $A_s$ | [m$^2$] | 0.423 | 1.350 | 0.674 |
| $\lambda$ | [1/s] | $1.92 \times 10^{-5}$ | $2.44 \times 10^{-5}$ | $1.02 \times 10^{-5}$ |
| $\lambda_S$ | [1/s] | $2.13 \times 10^{-5}$ | $1.83 \times 10^{-5}$ | $1.03 \times 10^{-5}$ |
| $\hat{\lambda}$ | [1/s] | $2.87 \times 10^{-4}$ | $1.27 \times 10^{-5}$ | $1.28 \times 10^{-5}$ |
| $\hat{\lambda}_S$ | [1/s] | $2.50 \times 10^{-11}$ | $2.50 \times 10^{-11}$ | $2.50 \times 10^{-11}$ |
| $K_d$ | [m$^3$/kg] | 200.000 | 200.000 | 200.000 |
| $\rho$ | [kg/m$^3$] | $8.98 \times 10^{-4}$ | $7.90 \times 10^{-3}$ | $4.14 \times 10^{-3}$ |
| $A_s/(A + A_s)$ | [-] | 0.259 | 0.295 | 0.325 |
| $P_D$ | [-] | 0.012 | 0.033 | 0.006 |
| $d_S$ | [m] | 0.212 | 0.338 | 0.169 |
| $L_s = u/\alpha$ | [m] | $1.34 \times 10^4$ | $5.17 \times 10^4$ | $1.50 \times 10^5$ |
| $T_{sto} = A_s/(\alpha A)$ | [h] | 4.8 | 9.5 | 17.7 |
| $T_{str} = 1/\alpha$ | [h] | 32 | 300 | 296 |
| $q_s = \alpha \times A$ | [m$^2$/s] | $1.06 \times 10^{-5}$ | $2.98 \times 10^{-6}$ | $1.32 \times 10^{-6}$ |
| $R_h = T_{sto}/L_s$ | [s/m] | 0.691 | 0.778 | 0.376 |
| $t^m_{mean} = Lu + 2D/u^2$ | [h] | 1.62 | 3.52 | 2.10 |
| $t^s_{mean} = (A_S/A) \times t^m_{mean}$ | [h] | 0.68 | 4.75 | 1.41 |
| $t_{mean} = t^m_{mean} + t^s_{mean}$ | [h] | 2.30 | 8.27 | 3.51 |
| $F^{200}_{med}$ | [-] | $4.13 \times 10^{-3}$ | $1.31 \times 10^{-3}$ | $4.93 \times 10^{-4}$ |

## 4. Conclusions

The [32]P BTCs obtained for unfiltered stream water samples provided a consistent picture of the retention and transport of a pulse input of phosphorus in the 2.6 km long stream reach comprising stretches of diverse hydromorphological conditions. The overall retention of phosphorus tracer in the studied stream reach in the timescale of 6 days was 46%, with the least efficient retention in the stretch differing from others by higher water velocity, coarser bottom sediments and less developed surface storage. The OTIS-P model of transient storage provided reasonable fits to in-stream BTCs of the simultaneously injected [3]H and [32]P tracers, while failed at reproducing the hyporheic BTCs that significantly varied with the depth in sediment. Metrics of transient storage derived from the OTIS-P model parameters reflect a small effect of transient storage on solute transport. The artificially straightened channel, low stream gradient and bottom sediments dominated by silt and fine sand limit the hyporheic exchange. The higher influence of transient storage in the upper stretch was due to the enhanced surface storage. The beaver dam impoundment showed no positive influence on transient storage and phosphorus retention relative to the upstream free-flowing part of the stream.

**Author Contributions:** D.Z. and P.W. contributed equally to this work. Both authors have read and agreed to the published version of the manuscript.

**Funding:** This work was carried out in the BONUS SOILS2SEA project, which received funding from BONUS (Art 185), funded jointly by the EU and Innovation Fund Denmark, The Swedish Environmental Protection Agency, The Polish National Centre for 390 Research and Development, The German Ministry for Education and Research, and The 391 Russian Foundation for Basic Research (RFBR).

**Institutional Review Board Statement:** Not applicable.

**Informed Consent Statement:** Not applicable.

**Data Availability Statement:** Not applicable.

**Acknowledgments:** This work could not be be carried out without the help in sample collection provided by Dominika Bar-Michalczyk, Hao Duong Van, Paweł Jodłowski and Tomasz Michalczyk. Paweł Jodłowski provided radiation protection support. The authors also thank members of the Częstochowa Division of the Polish Angling Association (PZW) for their kind approval of the tracer test.

**Conflicts of Interest:** The authors declare no conflict of interest.

## Abbreviations

The following abbreviations are used in this manuscript:

BTC     Breakthrough curve
TDP     Total dissolved phosphorus

## Appendix A. In-Stream and Hyporheic Tracer Concentrations

**Table A1.** In-stream $^3$H activity concentrations and their analytical uncertainties at points R1 and R2.

| | R1 | | | R2 | |
|---|---|---|---|---|---|
| Time [h] | c [kBq/L] | U(c) [kBq/L] | Time [h] | c [kBq/L] | U(c) [kBq/L] |
| 0.00 | 0.001 | 0.001 | 0.32 | 0.004 | 0.001 |
| 0.08 | 0.002 | 0.000 | 0.58 | 0.001 | 0.001 |
| 0.17 | 0.002 | 0.001 | 0.75 | 0.002 | 0.002 |
| 0.22 | 0.001 | 0.000 | 0.87 | 0.001 | 0.001 |
| 0.25 | 0.112 | 0.004 | 0.97 | 0.001 | 0.001 |
| 0.28 | 10.12 | 0.09 | 1.07 | 0.002 | 0.002 |
| 0.32 | 109.2 | 0.9 | 1.17 | 0.002 | 0.001 |
| 0.35 | 249.6 | 2.2 | 1.27 | 0.003 | 0.003 |
| 0.38 | 359.3 | 3.6 | 1.37 | 0.017 | 0.002 |
| 0.42 | 336.1 | 2.5 | 1.70 | 39.3 | 0.3 |
| 0.45 | 275.6 | 3.3 | 1.93 | 98.8 | 1.0 |
| 0.50 | 179.1 | 1.9 | 2.03 | 100.1 | 1.1 |
| 0.55 | 103.0 | 0.8 | 2.18 | 76.4 | 0.7 |
| 0.60 | 72.2 | 0.6 | 2.28 | 59.5 | 0.6 |
| 0.65 | 41.8 | 0.4 | 2.40 | 43.9 | 0.3 |
| 0.70 | 25.9 | 0.2 | 2.53 | 30.6 | 0.2 |
| 0.75 | 21.9 | 0.2 | 2.73 | 17.1 | 0.1 |
| 0.82 | 18.6 | 0.2 | 2.87 | 12.09 | 0.08 |
| 0.93 | 7.29 | 0.06 | 3.00 | 8.96 | 0.08 |
| 1.00 | 5.33 | 0.04 | 3.15 | 6.24 | 0.05 |
| 1.07 | 3.02 | 0.02 | 3.28 | 4.75 | 0.04 |
| 1.17 | 2.61 | 0.03 | 3.45 | 3.39 | 0.03 |
| 1.27 | 1.65 | 0.02 | 3.62 | 2.53 | 0.02 |
| 1.37 | 1.23 | 0.01 | 3.78 | 1.84 | 0.02 |
| 1.47 | 1.05 | 0.02 | 4.03 | 1.39 | 0.01 |
| 1.80 | 0.481 | 0.008 | 4.28 | 1.01 | 0.02 |
| 1.97 | 0.350 | 0.006 | 4.53 | 0.80 | 0.01 |
| 2.13 | 0.254 | 0.004 | 4.78 | 0.617 | 0.009 |
| 2.30 | 0.230 | 0.002 | 5.03 | 0.512 | 0.005 |
| 2.47 | 0.142 | 0.002 | 5.77 | 0.292 | 0.004 |
| 2.63 | 0.113 | 0.002 | 8.12 | 0.111 | 0.004 |
| 3.53 | 0.074 | 0.002 | 13.13 | 0.037 | 0.001 |
| 4.03 | 0.056 | 0.001 | 16.37 | 0.023 | 0.002 |
| 4.53 | 0.046 | 0.002 | 19.25 | 0.017 | 0.001 |
| 5.13 | 0.037 | 0.003 | 27.07 | 0.010 | 0.001 |
| 5.97 | 0.028 | 0.002 | 36.23 | 0.006 | 0.002 |
| 8.82 | 0.015 | 0.001 | 45.52 | 0.003 | 0.001 |
| 12.95 | 0.009 | 0.002 | 54.80 | 0.004 | 0.001 |
| 16.13 | 0.006 | 0.001 | 66.60 | 0.004 | 0.002 |
| 19.03 | 0.006 | 0.002 | 76.60 | 0.002 | 0.001 |
| 26.60 | 0.003 | 0.001 | 99.85 | 0.002 | 0.002 |
| 35.80 | 0.003 | 0.001 | 148.47 | 0.002 | 0.002 |
| 45.00 | 0.003 | 0.002 | | | |
| 52.25 | 0.003 | 0.001 | | | |
| 64.25 | 0.002 | 0.001 | | | |
| 121.25 | 0.001 | 0.001 | | | |
| 146.25 | 0.001 | 0.001 | | | |

**Table A2.** In-stream $^{3}H$ activity concentrations and their analytical uncertainties at points R3 and R4.

| R3 | | | R4 | | |
|---|---|---|---|---|---|
| Time [h] | c [kBq/L] | U(c) [kBq/L] | Time [h] | c [kBq/L] | U(c) [kBq/L] |
| 0.90 | 0.001 | 0.001 | 0.95 | 0.001 | 0.002 |
| 1.65 | 0.002 | 0.001 | 1.70 | 0.002 | 0.003 |
| 2.65 | 0.001 | 0.002 | 2.45 | 0.002 | 0.003 |
| 3.40 | 0.006 | 0.001 | 3.20 | 0.002 | 0.002 |
| 4.15 | 6.7 | 0.1 | 3.95 | 0.001 | 0.002 |
| 4.90 | 28.0 | 0.2 | 4.70 | 0.002 | 0.001 |
| 5.62 | 26.7 | 0.2 | 5.42 | 0.002 | 0.003 |
| 6.93 | 11.0 | 0.1 | 6.45 | 3.49 | 0.02 |
| 7.95 | 5.18 | 0.05 | 7.48 | 19.9 | 0.1 |
| 8.30 | 3.99 | 0.03 | 8.50 | 16.9 | 0.1 |
| 9.05 | 1.88 | 0.01 | 9.32 | 10.3 | 0.1 |
| 10.37 | 1.13 | 0.01 | 10.57 | 4.35 | 0.03 |
| 11.52 | 0.660 | 0.008 | 11.75 | 1.944 | 0.013 |
| 12.52 | 0.434 | 0.007 | 13.32 | 0.867 | 0.007 |
| 13.62 | 0.283 | 0.005 | 14.47 | 0.523 | 0.006 |
| 14.75 | 0.207 | 0.004 | 15.55 | 0.350 | 0.005 |
| 15.78 | 0.159 | 0.004 | 16.47 | 0.260 | 0.004 |
| 16.72 | 0.128 | 0.003 | 17.28 | 0.206 | 0.005 |
| 17.48 | 0.110 | 0.002 | 18.57 | 0.146 | 0.004 |
| 18.40 | 0.093 | 0.003 | 19.33 | 0.123 | 0.003 |
| 20.23 | 0.068 | 0.002 | 21.52 | 0.083 | 0.002 |
| 21.93 | 0.052 | 0.002 | 23.47 | 0.064 | 0.004 |
| 23.70 | 0.042 | 0.002 | 25.90 | 0.047 | 0.002 |
| 26.27 | 0.045 | 0.003 | 28.40 | 0.044 | 0.002 |
| 28.65 | 0.032 | 0.001 | 30.20 | 0.038 | 0.003 |
| 29.95 | 0.028 | 0.001 | 32.40 | 0.031 | 0.002 |
| 32.23 | 0.020 | 0.001 | 34.50 | 0.022 | 0.002 |
| 34.28 | 0.016 | 0.002 | 36.48 | 0.016 | 0.002 |
| 36.28 | 0.016 | 0.002 | 38.65 | 0.015 | 0.002 |
| 38.30 | 0.011 | 0.002 | 40.75 | 0.013 | 0.001 |
| 40.43 | 0.012 | 0.001 | 42.65 | 0.013 | 0.002 |
| 42.43 | 0.009 | 0.001 | 44.80 | 0.012 | 0.001 |
| 44.50 | 0.006 | 0.002 | 46.75 | 0.009 | 0.002 |
| 46.37 | 0.006 | 0.001 | 48.92 | 0.011 | 0.001 |
| 48.70 | 0.008 | 0.001 | 50.75 | 0.010 | 0.001 |
| 50.58 | 0.008 | 0.001 | 52.72 | 0.009 | 0.001 |
| 52.45 | 0.009 | 0.001 | 55.22 | 0.010 | 0.001 |
| 55.00 | 0.007 | 0.001 | 59.55 | 0.007 | 0.001 |
| 59.25 | 0.007 | 0.001 | 64.78 | 0.006 | 0.001 |
| 64.52 | 0.005 | 0.001 | 69.30 | 0.004 | 0.001 |
| 69.00 | 0.004 | 0.001 | 75.00 | 0.004 | 0.001 |
| 74.57 | 0.008 | 0.001 | 122.40 | 0.003 | 0.001 |
| 121.98 | 0.001 | 0.001 | 132.83 | 0.003 | 0.001 |
| 132.62 | 0.003 | 0.001 | 146.87 | 0.001 | 0.001 |
| 146.48 | 0.003 | 0.001 | | | |

**Table A3.** Hyporheic $^3$H activity concentrations and their analytical uncertainties at point R3.

| | R3 (5 cm) | | R3 (10 cm) | | R3 (15 cm) | |
|---|---|---|---|---|---|---|
| Time [h] | c [kBq/L] | U(c) [kBq/L] | c [kBq/L] | U(c) [kBq/L] | c [kBq/L] | U(c) [kBq/L] |
| 2.75 | 0.002 | 0.002 | 0.004 | 0.001 | 0.001 | 0.001 |
| 7.20 | 4.46 | 0.05 | 6.07 | 0.08 | 2.27 | 0.03 |
| 9.12 | 4.92 | 0.10 | 8.56 | 0.07 | 5.99 | 0.08 |
| 12.30 | 2.93 | 0.07 | 5.63 | 0.05 | 3.56 | 0.04 |
| 15.30 | 1.10 | 0.01 | 2.79 | 0.03 | 2.04 | 0.04 |
| 18.30 | 0.532 | 0.008 | 1.46 | 0.03 | 0.958 | 0.009 |
| 21.80 | 0.244 | 0.004 | 0.825 | 0.007 | 0.512 | 0.005 |
| 26.22 | 0.67 | 0.01 | 0.424 | 0.009 | 0.303 | 0.005 |
| 29.92 | 0.170 | 0.003 | 0.252 | 0.003 | 0.213 | 0.004 |
| 34.22 | 0.096 | 0.003 | 0.173 | 0.003 | 0.127 | 0.002 |
| 44.43 | 0.047 | 0.002 | 0.105 | 0.002 | 0.105 | 0.002 |
| 48.63 | 0.053 | 0.002 | 0.059 | 0.002 | 0.063 | 0.002 |
| 52.38 | 0.046 | 0.002 | 0.055 | 0.002 | 0.053 | 0.002 |
| 59.13 | 0.023 | 0.002 | 0.051 | 0.001 | 0.070 | 0.002 |
| 64.32 | 0.032 | 0.002 | 0.036 | 0.001 | 0.047 | 0.001 |
| 68.97 | 0.033 | 0.002 | 0.025 | 0.001 | 0.046 | 0.002 |
| 74.55 | 0.016 | 0.001 | 0.029 | 0.001 | 0.054 | 0.002 |
| 121.88 | 0.010 | 0.001 | 0.024 | 0.001 | 0.058 | 0.002 |
| 146.88 | 0.015 | 0.001 | 0.005 | 0.001 | 0.012 | 0.001 |

**Table A4.** Hyporheic $^3$H activity concentrations and their analytical uncertainties at point R4.

| | R4 (5 cm) | | R4 (10 cm) | |
|---|---|---|---|---|
| Time [h] | c [kBq/L] | U(c) [kBq/L] | c [kBq/L] | U(c) [kBq/L] |
| 3.00 | 0.002 | 0.001 | 0.001 | 0.001 |
| 5.00 | 0.005 | 0.001 | 0.001 | 0.001 |
| 7.58 | 6.1 | 0.1 | 0.281 | 0.002 |
| 9.33 | 12.0 | 0.1 | 1.72 | 0.01 |
| 11.92 | 6.03 | 0.07 | 4.31 | 0.04 |
| 14.92 | 2.28 | 0.02 | 4.77 | 0.02 |
| 18.03 | 0.748 | 0.008 | 2.64 | 0.02 |
| 21.48 | 0.279 | 0.003 | 1.67 | 0.01 |
| 25.98 | 0.111 | 0.002 | 1.099 | 0.008 |
| 30.18 | 0.067 | 0.002 | 0.792 | 0.005 |
| 34.48 | 0.041 | 0.001 | 0.464 | 0.004 |
| 38.65 | 0.022 | 0.002 | 0.315 | 0.002 |
| 44.78 | 0.020 | 0.001 | 0.160 | 0.001 |
| 48.90 | 0.019 | 0.001 | 0.151 | 0.001 |
| 52.70 | 0.013 | 0.001 | 0.135 | 0.001 |
| 59.52 | 0.009 | 0.001 | 0.063 | 0.001 |
| 64.92 | 0.028 | 0.001 | 0.078 | 0.001 |
| 69.15 | 0.035 | 0.001 | 0.076 | 0.001 |
| 74.48 | 0.024 | 0.001 | 0.075 | 0.001 |
| 122.48 | 0.033 | 0.002 | 0.175 | 0.002 |
| 146.48 | 0.007 | 0.001 | 0.119 | 0.001 |

**Table A5.** In-stream $^{32}$P activity concentrations and their analytical uncertainties at points R1 and R2.

| R1 | | | R2 | | |
|---|---|---|---|---|---|
| Time [h] | c [kBq/L] | U(c) [kBq/L] | Time [h] | c [kBq/L] | U(c) [kBq/L] |
| 0.00 | <LOQ | - | 0.32 | <LOQ | - |
| 0.08 | <LOQ | - | 0.58 | <LOQ | - |
| 0.17 | <LOQ | - | 0.75 | <LOQ | - |
| 0.22 | <LOQ | - | 0.87 | <LOQ | - |
| 0.25 | 0.03 | 0.00 | 0.97 | <LOQ | - |
| 0.28 | 3.06 | 0.03 | 1.07 | <LOQ | - |
| 0.32 | 31.8 | 0.3 | 1.17 | 0.010 | 0.007 |
| 0.35 | 70.6 | 0.6 | 1.27 | 0.010 | 0.007 |
| 0.38 | 103.0 | 0.9 | 1.37 | 0.01 | 0.02 |
| 0.42 | 96.2 | 0.9 | 1.70 | 7.9 | 0.1 |
| 0.45 | 75.9 | 0.7 | 1.93 | 19.0 | 0.2 |
| 0.50 | 47.9 | 0.5 | 2.03 | 18.0 | 0.2 |
| 0.55 | 26.8 | 0.3 | 2.18 | 13.6 | 0.2 |
| 0.60 | 18.0 | 0.2 | 2.28 | 10.5 | 0.1 |
| 0.65 | 10.6 | 0.1 | 2.40 | 7.6 | 0.1 |
| 0.70 | 6.46 | 0.09 | 2.53 | 5.4 | 0.1 |
| 0.75 | 5.60 | 0.08 | 2.73 | 3.42 | 0.06 |
| 0.82 | 4.73 | 0.07 | 2.87 | 2.61 | 0.05 |
| 0.93 | 1.96 | 0.04 | 3.00 | 2.08 | 0.05 |
| 1.00 | 1.45 | 0.03 | 3.15 | 1.63 | 0.03 |
| 1.07 | 1.02 | 0.03 | 3.28 | 1.39 | 0.03 |
| 1.17 | 0.82 | 0.03 | 3.45 | 1.19 | 0.03 |
| 1.27 | 0.68 | 0.03 | 3.62 | 1.07 | 0.03 |
| 1.37 | 0.51 | 0.01 | 3.78 | 0.91 | 0.02 |
| 1.47 | 0.439 | 0.008 | 4.03 | 0.74 | 0.01 |
| 1.80 | 0.301 | 0.007 | 4.28 | 0.63 | 0.01 |
| 1.97 | 0.265 | 0.006 | 4.53 | 0.57 | 0.01 |
| 2.13 | 0.238 | 0.006 | 4.78 | 0.55 | 0.01 |
| 2.30 | 0.197 | 0.005 | 5.03 | 0.452 | 0.009 |
| 2.47 | 0.149 | 0.004 | 5.77 | 0.352 | 0.008 |
| 2.63 | 0.129 | 0.004 | 8.12 | 0.180 | 0.006 |
| 3.53 | 0.098 | 0.003 | 13.13 | 0.076 | 0.004 |
| 4.03 | 0.070 | 0.004 | 16.37 | 0.067 | 0.004 |
| 4.53 | 0.070 | 0.004 | 19.25 | 0.041 | 0.004 |
| 5.13 | 0.050 | 0.004 | 27.07 | 0.042 | 0.003 |
| 5.97 | 0.052 | 0.005 | 36.23 | 0.018 | 0.003 |
| 8.82 | 0.030 | 0.003 | 45.52 | 0.013 | 0.003 |
| 12.95 | 0.017 | 0.004 | 54.80 | 0.011 | 0.003 |
| 16.13 | 0.017 | 0.004 | 66.60 | 0.015 | 0.001 |
| 19.03 | 0.010 | 0.004 | 76.60 | 0.012 | 0.001 |
| 26.60 | 0.005 | 0.004 | 99.85 | 0.004 | 0.001 |
| 35.80 | 0.011 | 0.001 | 148.47 | 0.002 | 0.001 |
| 45.00 | 0.011 | 0.001 | | | |
| 52.25 | 0.010 | 0.001 | | | |
| 64.25 | 0.007 | 0.001 | | | |
| 121.25 | 0.004 | 0.001 | | | |
| 146.25 | 0.002 | 0.001 | | | |



**Table A6.** In-stream $^{32}$P activity concentrations and their analytical uncertainties at points R3 and R4.

| R3 | | | R4 | | |
|---|---|---|---|---|---|
| Time [h] | c [kBq/L] | U(c) [kBq/L] | Time [h] | c [kBq/L] | U(c) [kBq/L] |
| 0.90 | <LOQ | - | 0.95 | <LOQ | - |
| 1.65 | <LOQ | - | 1.70 | <LOQ | - |
| 2.65 | <LOQ | - | 2.45 | <LOQ | - |
| 3.4000 | 0.001 | 0.005 | 3.20 | <LOQ | - |
| 4.1500 | 1.04 | 0.02 | 3.95 | <LOQ | - |
| 4.9000 | 3.64 | 0.05 | 4.70 | <LOQ | - |
| 5.6167 | 3.26 | 0.05 | 5.42 | <LOQ | - |
| 6.9333 | 1.39 | 0.02 | 6.45 | 0.37 | 0.02 |
| 7.9500 | 0.80 | 0.02 | 7.48 | 1.87 | 0.02 |
| 8.3000 | 0.67 | 0.02 | 8.50 | 1.64 | 0.02 |
| 9.0500 | 0.51 | 0.01 | 9.32 | 1.02 | 0.01 |
| 10.3667 | 0.330 | 0.009 | 10.57 | 0.568 | 0.009 |
| 11.5167 | 0.225 | 0.009 | 11.75 | 0.364 | 0.009 |
| 12.5167 | 0.212 | 0.007 | 13.32 | 0.245 | 0.007 |
| 13.6167 | 0.178 | 0.005 | 14.47 | 0.200 | 0.005 |
| 14.7500 | 0.155 | 0.005 | 15.55 | 0.157 | 0.005 |
| 15.7833 | 0.119 | 0.006 | 16.47 | 0.137 | 0.006 |
| 16.7167 | 0.118 | 0.006 | 17.28 | 0.126 | 0.006 |
| 17.4833 | 0.105 | 0.006 | 18.57 | 0.103 | 0.006 |
| 18.4000 | 0.092 | 0.006 | 19.33 | 0.115 | 0.006 |
| 20.2333 | 0.089 | 0.004 | 21.52 | 0.100 | 0.004 |
| 21.9333 | 0.078 | 0.004 | 23.47 | 0.070 | 0.004 |
| 23.7000 | 0.075 | 0.004 | 25.90 | 0.066 | 0.004 |
| 26.2667 | 0.073 | 0.005 | 28.40 | 0.068 | 0.005 |
| 28.6500 | 0.067 | 0.005 | 30.20 | 0.056 | 0.005 |
| 29.9500 | 0.067 | 0.005 | 32.40 | 0.044 | 0.005 |
| 32.2333 | 0.062 | 0.005 | 34.50 | 0.048 | 0.005 |
| 34.2833 | 0.059 | 0.006 | 36.48 | 0.038 | 0.006 |
| 36.2833 | 0.048 | 0.006 | 38.65 | 0.037 | 0.006 |
| 38.3000 | 0.034 | 0.006 | 40.75 | 0.038 | 0.006 |
| 52.4500 | 0.029 | 0.004 | 42.65 | 0.013 | 0.005 |
| 55.0000 | 0.024 | 0.003 | 44.80 | 0.020 | 0.005 |
| 59.2500 | 0.028 | 0.003 | 46.75 | 0.009 | 0.005 |
| 64.5167 | 0.020 | 0.003 | 48.92 | 0.026 | 0.009 |
| 69.0000 | 0.019 | 0.001 | 50.75 | 0.027 | 0.006 |
| 74.5667 | 0.021 | 0.001 | 52.72 | 0.027 | 0.010 |
| 121.9833 | 0.005 | 0.001 | 55.22 | 0.026 | 0.004 |
| 132.6167 | 0.005 | 0.001 | 59.55 | 0.018 | 0.003 |
| 146.4833 | 0.002 | 0.001 | 64.78 | 0.025 | 0.003 |
| | | | 69.30 | 0.021 | 0.003 |
| | | | 75.00 | 0.019 | 0.001 |
| | | | 122.40 | 0.005 | 0.001 |
| | | | 132.83 | 0.005 | 0.001 |
| | | | 146.87 | 0.003 | 0.001 |

**Table A7.** Hyporheic $^{32}$P activity concentrations and their analytical uncertainties at point R3.

| | R3 (5 cm) | | R3 (10 cm) | | R3 (15 cm) | |
|---|---|---|---|---|---|---|
| Time [h] | c [kBq/L] | U(c) [kBq/L] | c [kBq/L] | U(c) [kBq/L] | c [kBq/L] | U(c) [kBq/L] |
| 2.75 | <LOQ | - | <LOQ | - | <LOQ | - |
| 7.20 | 0.119 | 0.003 | 0.036 | 0.002 | 0.017 | 0.001 |
| 9.12 | 0.088 | 0.003 | 0.044 | 0.002 | 0.024 | 0.001 |
| 12.30 | 0.042 | 0.002 | 0.014 | 0.002 | 0.008 | 0.001 |
| 15.30 | 0.028 | 0.002 | 0.003 | 0.003 | 0.003 | 0.001 |
| 18.30 | 0.020 | 0.002 | 0.002 | 0.003 | 0.002 | 0.001 |
| 21.80 | 0.008 | 0.003 | 0.001 | 0.003 | 0.002 | 0.001 |
| 26.22 | 0.012 | 0.002 | 0.001 | 0.003 | 0.001 | 0.002 |
| 29.92 | 0.009 | 0.002 | <LOQ | - | <LOQ | - |
| 34.22 | 0.008 | 0.002 | 0.002 | 0.002 | 0.001 | 0.002 |
| 44.43 | 0.010 | 0.001 | 0.003 | 0.001 | 0.008 | 0.001 |
| 48.63 | 0.014 | 0.001 | 0.009 | 0.001 | 0.004 | 0.002 |
| 52.38 | 0.013 | 0.001 | 0.006 | 0.001 | 0.006 | 0.001 |
| 59.13 | 0.011 | 0.001 | 0.006 | 0.001 | 0.004 | 0.001 |
| 64.32 | 0.006 | 0.001 | 0.007 | 0.001 | 0.003 | 0.001 |
| 68.97 | 0.010 | 0.001 | 0.005 | 0.001 | 0.003 | 0.001 |
| 74.55 | 0.004 | 0.001 | <LOQ | - | 0.002 | 0.001 |
| 121.88 | <LOQ | - | 0.010 | 0.001 | 0.001 | 0.001 |
| 146.88 | <LOQ | - | 0.001 | 0.001 | <LOQ | - |

**Table A8.** Hyporheic $^{32}$P activity concentrations and their analytical uncertainties at point R4.

| | R4 (5 cm) | | R4 (10 cm) | |
|---|---|---|---|---|
| Time [h] | c [kBq/L] | U(c) [kBq/L] | c [kBq/L] | U(c) [kBq/L] |
| 3.00 | <LOQ | - | <LOQ | - |
| 5.00 | <LOQ | - | 0.011 | 0.002 |
| 7.58 | 0.140 | 0.008 | 0.014 | 0.002 |
| 9.33 | 0.182 | 0.010 | 0.023 | 0.002 |
| 11.92 | 0.092 | 0.005 | 0.019 | 0.002 |
| 14.92 | 0.056 | 0.003 | 0.013 | 0.002 |
| 18.03 | 0.036 | 0.003 | 0.008 | 0.002 |
| 21.48 | 0.031 | 0.002 | 0.008 | 0.002 |
| 25.98 | 0.027 | 0.002 | 0.003 | 0.002 |
| 30.18 | 0.022 | 0.002 | 0.007 | 0.002 |
| 34.48 | 0.025 | 0.002 | 0.010 | 0.001 |
| 38.65 | 0.023 | 0.002 | 0.011 | 0.001 |
| 44.78 | 0.009 | 0.001 | 0.001 | 0.001 |
| 48.90 | 0.011 | 0.001 | 0.001 | 0.001 |
| 52.70 | 0.011 | 0.001 | 0.002 | 0.001 |
| 59.52 | 0.013 | 0.001 | 0.003 | 0.001 |
| 64.92 | 0.008 | 0.001 | 0.002 | 0.001 |
| 69.15 | 0.008 | 0.001 | 0.003 | 0.001 |
| 74.48 | 0.009 | 0.001 | 0.003 | 0.001 |
| 122.48 | 0.003 | 0.001 | 0.007 | 0.001 |
| 146.48 | <LOQ | - | <LOQ | - |

**Table A8.** *Cont.*

| | R4 (5 cm) | | R4 (10 cm) | |
|---|---|---|---|---|
| **Time [h]** | **c [kBq/L]** | **U(c) [kBq/L]** | **c [kBq/L]** | **U(c) [kBq/L]** |
| 3.00 | <LOQ | - | <LOQ | - |
| 5.00 | <LOQ | - | 0.011 | 0.002 |
| 7.58 | 0.140 | 0.008 | 0.014 | 0.002 |
| 9.33 | 0.182 | 0.010 | 0.023 | 0.002 |
| 11.92 | 0.092 | 0.005 | 0.019 | 0.002 |
| 14.92 | 0.056 | 0.003 | 0.013 | 0.002 |
| 18.03 | 0.036 | 0.003 | 0.008 | 0.002 |
| 21.48 | 0.031 | 0.002 | 0.008 | 0.002 |
| 25.98 | 0.027 | 0.002 | 0.003 | 0.002 |
| 30.18 | 0.022 | 0.002 | 0.007 | 0.002 |
| 34.48 | 0.025 | 0.002 | 0.010 | 0.001 |
| 38.65 | 0.023 | 0.002 | 0.011 | 0.001 |
| 44.78 | 0.009 | 0.001 | 0.001 | 0.001 |
| 48.90 | 0.011 | 0.001 | 0.001 | 0.001 |
| 52.70 | 0.011 | 0.001 | 0.002 | 0.001 |
| 59.52 | 0.013 | 0.001 | 0.003 | 0.001 |
| 64.92 | 0.008 | 0.001 | 0.002 | 0.001 |
| 69.15 | 0.008 | 0.001 | 0.003 | 0.001 |
| 74.48 | 0.009 | 0.001 | 0.003 | 0.001 |
| 122.48 | 0.003 | 0.001 | 0.007 | 0.001 |
| 146.48 | <LOQ | - | <LOQ | - |

## Appendix B.

Variables and parameters of the OTIS-P model (after [44]).

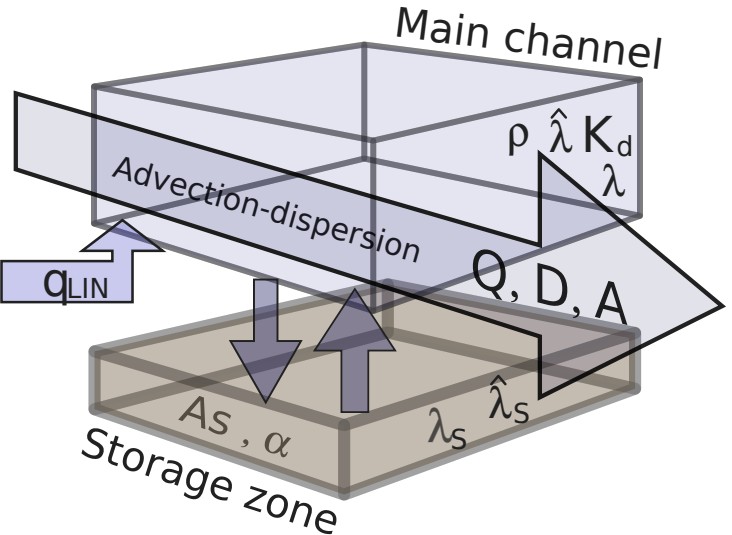

**Figure A1.** Conceptual mode of OTIS-P. Symbols explained in Appendix C.

### Appendix C. List of Symbols

| | |
|---|---|
| $C$ | main channel solute concentration |
| $C_L$ | lateral inflow solute concentration |
| $C_S$ | storage zone solute concentration |
| $L$ | reach length |
| $A$ | main channel cross-section area |
| $Q$ | volumetric flow rate (discharge) |
| $u$ | mean water velocity |
| $D$ | dispersion coefficient |
| $\alpha$ | storage zone exchange coefficient |
| $A_s$ | storage zone cross-section area |
| $\lambda$ | main channel first-order decay coefficient |
| $\lambda_S$ | storage zone first-order decay coefficient |
| $\hat{\lambda}$ | main channel sorption rate coefficient |
| $\hat{\lambda}_S$ | storage zone sorption rate coefficient |
| $K_d$ | distribution coefficient |
| $\rho$ | mass of accesible sediment/volume water |
| $P_D$ | dispersion parameter |
| $d_S$ | storage zone depth |
| $L_s$ | average distance a molecule travels downstream within the main channel prior to entering the storage zone |
| $T_{sto}$ | the main channel residence time |
| $T_{str}$ | is the storage zone residence time |
| $q_s$ | storage exchange flux |
| $R_h$ | hydrological retention factor |
| $t_{mean}^{m}$ | mean travel time due to the main channel |
| $t_{mean}^{s}$ | mean travel time due to storage zone |
| $t_{mean}$ | mean travel time |
| $F_{med}^{200}$ | fraction of median travel time due to transient storage |

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
