# Peer review of "Phosphorus Transport in a Lowland Stream Derived from a Tracer Test with 32P"

_water, doi:10.3390/w13081030_

Round 1

Reviewer 1 Report

Dear Authors,

please consider some minor revisions in the pdf file.

Best regards

Author Response

Dear Reviewer,

Thank you for your corrections. We have accepted all of them in the revised version of the manuscript.

Authors

Reviewer 2 Report

The paper deals with an important issue affecting surface water; the phosphorus enrichment in these waters can be a serious environmental problem in areas in which the anthropic impact is high.

More in specific, the paper addresses the P-transport in a lowland stream by means of tracer tests.

The introduction is well written, the use of P as a tracer is well argued and sufficiently developed, the main objectives of the paper are clearly indicated. Both the objectives are discussed together with the way they will be addressed.

The only issue needing a further develop and more detailed explanation is that regarding the use of radioactive phosphorus instead of the corresponding stable isotope; since in some countries radioactive tracer experiments are limited or forbidden, and therefore the use of such a tracer might be not so widely diffused, please indicate with more accuracy and detail the reasons why you suggest the use of 32P in these types of application.

Why is it preferable to stable isotope of phosphorus?

In the Materials and methods section, I suggest to indicate also the lithological nature of aquifers (Jurassic and Quaternary) feeding the stream, i.e. they are carbonatic or arenaceous, fissured or porous and so on.

The authors state that during the tracer test stream discharge was not constant along the stream length; you should discuss whether this fact has an impact on tracer test results (especially tritium tracer test) or not, and, in the event, quantify this impact.

Please give more information about the easy-soluble tracer test performed before the test.

Methods description is very accurate and ensures a good repeatability of the tests. I suggest to introduce the OTIS-P model in a more comprehensive way.

Results and discussion section contains a detailed description of the obtained results and permits a good reading of the discussion, especially in the characteristics of tracer transport, very in deep explained.

The assumptions and data interpretations are well supported in the discussion, I agree with the authors explanations. 

Some other minor suggestions are in the attached pdf file

Author Response

Dear Reviewer,

Thank you for your comments and corrections.

All minor comments provided in the pdf file were accepted and approppriate corrections were made.

Comments and suggestions

  1. Introduction - only issue needing a further develop and more detailed explanation is that regarding the use of radioactive phosphorus instead of the corresponding stable isotope; since in some countries radioactive tracer experiments are limited or forbidden, and therefore the use of such a tracer might be not so widely diffused, please indicate with more accuracy and detail the reasons why you suggest the use of 32P in these types of application. Why is it preferable to stable isotope of phosphorus?

The principal advantage of radioactive phosphorus additions is that a negligible amount phosphorus added to a stream does not affect phosphorus cycling. We have added following sentence to the Introduction.

“An important advantage of radioactive phosphorus injections, comparing to stable phosphorus additions, is that they introduce a negligible amount of phosphorus, which does not affect stream uptake capacity. “

The disadvantages of the use of the stable phosphorus additions in studies on phosphorus cycling in streams are described in the literature. They tended to overestimate the uptake lengths (Mulholland et al., 1990; Payn et al., 2005) and failed at providing the relative contributions of biota and abiotic processes in phosphorus uptake (Stutter et al., 2010). It was not our intention to suggest that the method based on the radioactive tracer is absolutely superior to stable phosphorus additions. We have rewritten the sentence on the recent developments of that method emphasizing the ability of the refined approaches to cope with those limitations.

“Consequently, methods based on stable phosphorus addition to quantify the relative contribution of abiotic and biotic processes [16] and to quantify uptake lengths [31] have been refined and methods based on tracing stable isotopic composition of oxygen in phosphate [32–34] have been developed.”

  1. I suggest to indicate also the lithological nature of aquifers (Jurassic and Quaternary) feeding the stream, i.e. they are carbonatic or arenaceous, fissured or porous and so on.

Information on the lithology of the aquiferes has been added.

  1. The authors state that during the tracer test stream discharge was not constant along the stream length; you should discuss whether this fact has an impact on tracer test results (especially tritium tracer test) or not, and, in the event, quantify this impact

Discharge changes during the experiment were twofold. Discharge increased downstream the studied stretch due to lateral inflows. Additionally, the decrease of discharge during the experiment was associated with the falling limb of the hydrograph that followed a precipitation event on 31 March//1 April. We guess that the information on discharge range from 100 to 200 l/s might be misunderstood. The discharge did not increase from 100 l/s to 200 l/s,  but varied within that range. This question was clarified in the text. Both the temporal and spatial variations of stream discharge (Table 1) were taken into account explicitly in the OTIS-P model (Tables 4 and 5), where discharge values were interpolated linearly for each spatial and temporal (at 1 h resolution) step.   

  1. Please give more information about the easy-soluble tracer test performed before the test.

Information on the previous test was added.

“.Therefore, the conservative tracer BTCs were predicted for the sampling points by the OTIS-P model (without transient storage) using discharge and water velocity values measured on the day preceding tracer injection and dispersion coefficients derived from an independent tracer experiment.  The BTCs obtained during the nitrate addition experiment performed in October 2017 in the stretch I - R1 (Q= 123 l/s) were used to estimate the dispersion coefficient.”

  1. Methods description is very accurate and ensures a good repeatability of the tests. I suggest to introduce the OTIS-P model in a more comprehensive way.

An extended subchapter 2.4 and Appendices B and C provide a description of OTIS-P and of the modeling procedure.

Mulholland, P.J., Steinman, A.D. and Elwood, J.W., 1990. Measurement of phosphorus uptake length in streams: comparison of radiotracer and stable PO4 releases. Canadian Journal of Fisheries and Aquatic Sciences, 47(12): 2351-2357.

Payn, R.A., Webster, J.R., Mulholland, P.J., Valett, H.M. and Dodds, W.K., 2005. Estimation of stream nutrient uptake from nutrient addition experiments. Limnology and Oceanography: Methods, 3(3): 174-182.

Stutter, M., Demars, B. and Langan, S., 2010. River phosphorus cycling: Separating biotic and abiotic uptake during short-term changes in sewage effluent loading. Water research, 44(15): 4425-4436.

Reviewer 3 Report

Dear Author,

this is an interesting manuscript about anthropogenic phosphorus enrichment in a river basin. My opinion is that it could be worthy of publication after major revisions. Some suggestions:

Introduction: row 30. It's necessary to add and sentence and the relative reference like: "The structure, status, and processes of the groundwater system, which can only be acquired through scientific research efforts, are critical aspects of water resource management. In this regard, stable and radioactive isotope data provide essential tools in support of water resources management" (Water 201911(2), 291; https://doi.org/10.3390/w11020291). 

Enlarge the discussion about the importance of this study, in particular for the agricultural landscape;

Methods

give quality control about the isotopic analyses;

The sampling was related to which period? What about the two sampling points? please specify.

There was a correlation between the sampling and climatic conditions? please add some information about the climatic conditions in the Introduction.

In table 1 you refer to 4 sampling point, while in the text (row 104) you report 2 single points...?

Conclusion

please stress more about the worldwide implications of this approach etc etc 

Evaluate to add as supplementary materials all the appendix for a more suitable reading of the manuscript.

Author Response

Dear rRviewer,

Thank you for the comments and corrections. Below you will find our responses to your remarks.

  1. Introduction: row 30. It's necessary to add and sentence and the relative reference like: "The structure, status, and processes of the groundwater system, which can only be acquired through scientific research efforts, are critical aspects of water resource management. In this regard, stable and radioactive isotope data provide essential tools in support of water resources management" (Water201911(2), 291; https://doi.org/10.3390/w11020291). 

The following sentence has been added with additional references:

Tracing of intentionally introduced substances (artificial tracers) at field and laboratory scales in order to understand the functioning and status of hydrological systems is an essential tool in water resources management [10–14].”

  1. Enlarge the discussion about the importance of this study, in particular for the agricultural landscape;

In the following sentences added to the text we  indicate the relevance of our results to the changes brought to small streams by the worldide phenomenon of urban sprawl.

The results of this study contribute to the understanding of the fate of the episodic inputs of inorganic phosphorus in channelized streams during the non-growing season. The relative importance of such episodic, semi-diffuse sources of phosphorus to small streams

are expected to increase due to urban sprawl [45].

  1. give quality control about the isotopic analyse

The quality of the analyses of the activities of tracers was controlled through the measurements of reference solutions for both isotopes. An additional sentence explains this.

The activities of the reference solutions were used to determine counting efficiency for each batch of samples.

  1. The sampling was related to which period? What about the two sampling points? please specify.

There was a correlation between the sampling and climatic conditions? please add some information about the climatic conditions in the Introduction.

In table 1 you refer to 4 sampling point, while in the text (row 104) you report 2 single points...?

The four points of Table 1 are the sampling points of the tracer experiment, while the two sampling points referred to in line 104 were used in the water quality monitoring programme conducted in the years 2014 – 2017.

An explanation on the weather conditions was added.

The weather was stable during the experiment, with no precipitation and air temperatures fluctuating between -2 to +10â—¦C

  1. please stress more about the worldwide implications of this approach etc etc

See point 2. above.

  1. Evaluate to add as supplementary materials all the appendix for a more suitable reading of the manuscript.

We are leaving this question to the decision of the Editor.

Round 2

Reviewer 3 Report

The authors followed my reviewer comments and now the paper is significantly improved and appropriate for publication.